# Efficient realization of quantum primitives for Shor's algorithm using PennyLane library

A. V. Antipov[1,2]*, E. O. Kiktenko[1,2], A. K. Fedorov[1,2]

1 Russian Quantum Center, Skolkovo, Moscow, Russia, 2 National University of Science and Technology "MISIS", Moscow, Russia

* an.antipov@rqc.ru, aantipov@nes.ru

## Abstract

Efficient realization of quantum algorithms is among main challenges on the way towards practical quantum computing. Various libraries and frameworks for quantum software engineering have been developed. Here we present a software package containing implementations of various quantum gates and well-known quantum algorithms using PennyLane library. Additoinally, we used a simplified technique for decomposition of algorithms into a set of gates which are native for trapped-ion quantum processor and realized this technique using PennyLane library. The decomposition is used to analyze resources required for an execution of Shor's algorithm on the level of native operations of trapped-ion quantum computer. Our original contribution is the derivation of coefficients needed for implementation of the decomposition. Templates within the package include all required elements from the quantum part of Shor's algorithm, specifically, efficient modular exponentiation and quantum Fourier transform that can be realized for an arbitrary number of qubits specified by a user. All the qubit operations are decomposed into elementary gates realized in PennyLane library. Templates from the developed package can be used as qubit-operations when defining a QNode.

## Introduction

The use of the laws of quantum mechanics could give rise to a new computing paradigm that is believed to be superior to classical computing for a certain class of problems [1]. Recent advances in the realization of quantum computing devices based on diverse physical principles, such as solid-state systems [2–4], trapped ions [5, 6], and neutral atoms [7, 8], have pushed their capabilities to the threshold of quantum advantage. In addition to progress in quantum hardware, software aspects of quantum computing attracted a significant deal of interest. Various libraries and frameworks for programming quantum devices have been suggested [9, 10]. Still, one of the most important aspects of their use is a sufficient amount of pre-programmed packages for quantum algorithms and their building blocks. With the increase of the complexity of quantum algorithms, well-tested packages for primary quantum primitives become of rising importance.

**Data Availability Statement:** All relevant data are within the paper and its Supporting information files.

**Funding:** This work is supported by the grant of the Russian Science Foundation No. 19-71-10091

(realization of quantum algorithms), Leading Research Center on Quantum Computing (Agreement 014/20; transpilation), and by the Priority 2030 program at the National University of Science and Technology MISIS (resource-estimation and implementation analysis).

**Competing interests:** The authors have declared that no competing interests exist.

One of existing software platforms is PennyLane, which is a cross-platform Python library for programming quantum computers. Its main application focuses on optimization tasks in quantum and hybrid quantum-classical algorithms. An interesting feature of PennyLane is that it is a unified architecture that can in principle be used with any gate-based or quantum computing platform or quantum simulator as a backend [11]. This feature makes PennyLane appealing for realizations of many well-known quantum algorithms that can be used, first, for demonstrative, educational, and research purposes, and, in future, for solving practical problems.

In this work, we present a set of functions that form the basis for the realization of Shor's algorithm [12] using PennyLane library. See the source at [13]. We realize functions that include all required elements from the quantum part of Shor's algorithm: efficient modular exponentiation and quantum Fourier transform. These important quantum primitives can be realized for an arbitrary number of qubits specified by a user. All qubit operations are decomposed into PennyLane's elementary gates. Functions from the package are realized as templates and can be used as qubit-operations when defining a QNode. We expect that our results are directly applicable for programming quantum devices using PennyLane library.

Realization of the mentioned algorithms allows for easy resource-estimation in the terms of quantum gates, because decompositions are explicitly defined inside these functions. We focused on ion-trapped quantum processor and developed functionality for transpilation of decompositions from given quantum algorithm into a set of native single- and two-qubit gates. Apart from reduction in noise levels due to the use of less noisy gates, the transpilation provides means for counting the amount of native gates and estimation of the execution time for a given algorithm.

This paper is organized as follows. In Sec. 1, we provide a general overlook of the package. Sec. 2 contains the description of the quantum order-finding procedure that is necessary for the realization of Shor's algorithm. The most important building block for efficient realization of the order-finding procedure is the quantum modular exponentiation, so we devote Sec. 3 to the description of the architecture of quantum modular exponentiation in the case of 3-bit integer inputs. Although the modular exponentiation procedure in our package is realized for arbitrary $n$-bit input, we chose 3-bit inputs for illustrative purposes. Sec. 4 contains an example of usage of order-finding procedure. Sec. 5 contains a description of transpilation technique. For an illustration of resource-estimation, we provide a table containing counts of native gates in the order-finding procedure and depth of the circuit.

## Materials and methods

In this work, we used the PennyLane library as a basis for developing a software package containing efficient realizations of building blocks for important quantum algorithms. The main source with a theoretical description of the realized decompositions is [14]. The PennyLane software package utilized in this work is described in [11].

Repository containing realization of the developed modules is provided in [13]. The protocol associated with this repository can be accessed via the link https://dx.doi.org/10.17504/protocols.io.b5qaq5se[PROTOCOL DOI] (see [15]).

## Results

## 1 General description

The developed package contains quantum circuits realized as PennyLane's templates and a class of classical functions for auxiliary computations. Every template provides a decomposition of

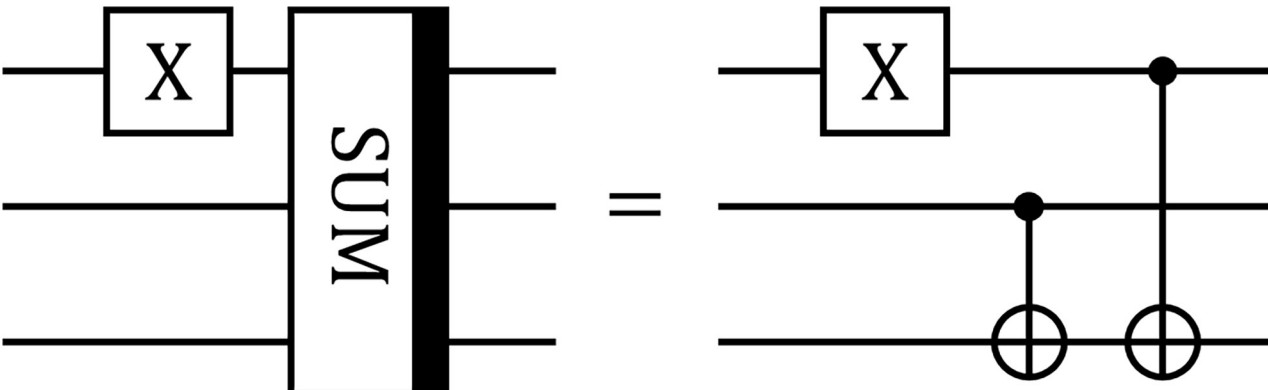

**Fig 1. Example of the gate SUM.** Quantum circuit contains one standard `PauliX` gate from the PennyLane library and one `SUM` gate from the list of added PennyLane templates.

desired qubit operation to the level of basic PennyLane gates and a class of classical functions helps to build some decompositions. The full list of templates and classical functions is provided in tables below.

We note that the developed templates can be used in the same way as elementary gates inside PennyLane's QNode structure. In Listing 1, the script implements the 3-qubit quantum circuit with 1 standard `PauliX` gate from the PennyLane library and the gate `SUM` that is realized as a template. The circuit is depicted in Fig 1.

**Listing 1.** Example of usage of the template `SUM`. Circuit in Fig 1 is realized.

```
import pennylane as qml
import QuantumOperations as q
# wires
wires = [0, 1, 2]
# device
dev = qml.device ('default.qubit', wires=wires shots=1000,
analytic=None)
# circuit
def func ():
  # use standard PennyLane's gate
  qml.PauliX (wires = wires [0])
  # use template SUM
  q.SUM(wires = wires)
  return qml.probs (wires)
# QNode
circuit = qml.QNode(func, dev)
```

1. **PennyLane's templates** with higher-level functions realizing quantum computations within PennyLane library are presented in Table 1.

2. **Functions from the class ClassicalOperations** for auxiliary computations are presented in Table 2.

The construction of the circuit is defined in the function func() and it is used to define the QNode object. QNode is a class that is used to construct quantum nodes encapsulating a quantum function or circuit and the computational device it is executed on.

Classical function `modular_multiplicative_inverse` is crucial for building the decomposition and it is used as the auxiliary function for `MODULAR_EXPONENTIATION`. The role of this classical function is to find parameters for the lower-level decomposition

**Table 1. PennyLane's templates developed for realization of quantum gates.**

| Gate | Description |
|---|---|
| SUM | Performs 3-qubit addition modulo 2 operation and puts the result in the third qubit |
| CARRY | Performs calculation of the highest order bit in the sum of three bits |
| CARRY_inv | Reversed (conjugate-transposed) CARRY gate |
| ADDER | Performs addition of two integer numbers encoded in input-qubits with respective binary representations |
| ADDER_inv | Reversed (conjugate-transposed) ADDER gate |
| ADDER_MOD | Performs addition modulo $N$ of two integer numbers $a$, $b < N$ encoded in input-qubits with respective binary representations |
| ADDER_MOD_inv | Reversed (conjugate-transposed) ADDER_MOD gate |
| Ctrl_MULT_MOD | If a control-qubit is $|1\rangle$, the gate performs multiplication of the integer number $z$ encoded in the input register by integer number $m$ modulo $N$; if the control-qubit is $|0\rangle$, then the initial number $z$ is put into the output register |
| Ctrl_MULT_MOD_inv | Reversed (conjugate-transposed) Ctrl_MULT_MOD gate |
| Ctrl_SWAP | Performs SWAP of two target-qubits conditional on the state of a control-qubit |
| MODULAR_EXPONENTIATION | Performs $O(n^3)$ modular exponentiation, in particular, for encoded into the input register integer number $x$, the gate performs calculation of $y^x$ modulo $N$ and puts the result into the output register |
| CR_k | Performs 2-qubit controlled phase shift gate which is used in the QFT (Quantum Fourier Transform) gate |
| CR_k_inv | Reversed (conjugate-transposed) CR_k gate |
| QFT_ | Performs Quantum Fourier Transform |
| QFT_inv | Performs reversed (conjugate-transposed) Quantum Fourier Transform |
| Order_Finding | Performs quantum order-finding algorithm |

**Table 2. Classical auxiliary functions.**

| Function | Description |
|---|---|
| gcd | Performs Euclid's algorithm for finding greater common divider (GCD) of integers $a$ and $b$ |
| diophantine_equation | Solves Diophantine equation, i.e. given $a$, $b$, the function returns $x$, $y$ such that $ax + by = \mathrm{GCD}(a, b)$ |
| modular_multiplicative_inverse | Finds modular multiplicative inverse of an integer $a$ modulo $N$ using the function diophantine_equation |

Ctrl_MULT_MOD_inv. Important aspect of modular_multiplicative_inverse is the efficiency of the realization that relies on the efficiency of two other classical functions gcd and diophantine_equation.

## 2 Order-finding circuit description

Order-finding is the only quantum part in Shor's algorithm for integer factorization [12]. The procedure of the reduction of integer factorization task to order-finding task is given in S1 Appendix.

The circuit for realization of order-finding procedure is given in Fig 2. It consists of three blocks: succession of Hadamard gates, modular exponentiation and conjugate transpose of quantum Fourier transform. The order-finding is realized as the template Order_Finding and it makes use of templates QFT_ and MODULAR_EXPONENTIATION.

**Listing 2.** Order_Finding template.

```
# O(n^3) efficient order-finding circuit
# input parameters: N,y
class Order_Finding (Operation):
  num_params = 3
  num_wires = AnyWires
  par_domain = None
  @staticmethod
  def decomposition(* parameters, wires):
```

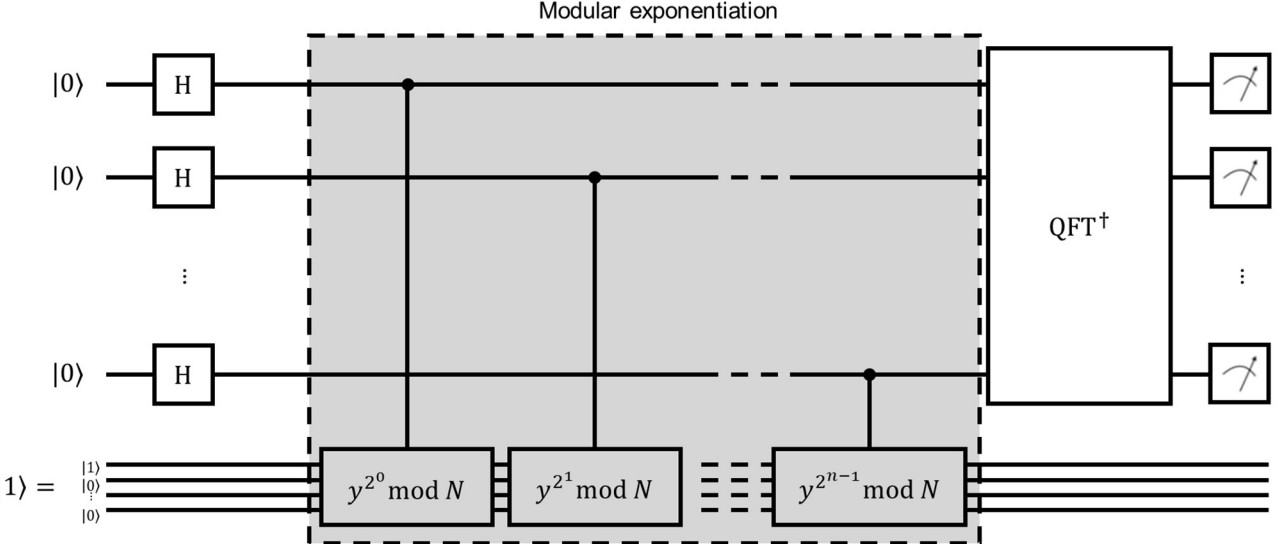

**Fig 2. Quantum circuit implementing the order-finding procedure.**

```python
# check wires and define registers
n_x = int (parameters [2])
if (len (wires)-2-n_x)%5 != 0:
  raise Exception('Wrong size of registers')
else:
  N = int (parameters [0])
  y = int (parameters [1])
  n = int ((len (wires)-2-n_x)/5)
  wires_x = wires [0:n_x]
  wires_z = wires [n_x:n_x+n]
  wires_a = wires [n_x+n:n_x+2*n]
  wires_b = wires [n_x+2*n:n_x+3*n+1]
  wires_c = wires [n_x+3*n+1:n_x+4*n+1]
  wires_N = wires [n_x+4*n+1:n_x+5*n+1]
  wires_t = wires [-1]
# check inputs
# check N
# check if N does not match the size of wires_N
if N > 2**(len (wires_N))-1:
  raise Exception ('N is too big')
with qml.tape.OperationRecorder() as rec:
  # Create superposition with Hadamard gates
  for i in range (len (wires_x)):
    qml.Hadamard(wires = wires_x[i])
  # Apply modular exponentiation
  MODULAR_EXPONENTIATION (N, y, n_x,\ wires = wires_x+wires_z+wir-
es_a+wires_b+\ wires_c+wires_N+[wires_t])
  # Apply inverse Quantum Fourier transform
  # to the first register
  QFT_inv (wires = wires_x)
  return rec.queue
```

Listing 2 demonstrates how Order_Finding is realized in the package. As one can see, there are only commands to add gates to the circuit. All other necessary elements, such as

initializing the circuit with a particular state and performing measurements at the end of the circuit, should be performed in the QNode environment in a similar fashion to Listing 1.

## 3 Modular exponentiation circuit description

Here we describe the architecture of efficient $O(n^3)$ modular exponentiation circuit from Ref. [14], for the case $n = 3$ using specific 3-bit numeric values in order to make the general approach more illustrative. The circuit is not subject to further lower-level optimization, but it is still efficient and replicates the logic behind the commonly used decomposition technique.

In general, modular exponentiation is the procedure of finding the value $y^x$ mod $N$ when $x$, $y$ and $N$ are given integers. The template MODULAR_EXPONENTIATION in Fig 3 is developed for solving this task.

Registers denoted by subscripts **x** and **N** should contain quantum states corresponding to binary representations of integers $x$ and $N$. Particular values of $y$ and $N$ define the architecture of the circuit. Register **z** should contain the binary representation of the solution $|y^x \bmod N\rangle$ at the end of the circuit, and it should be initialized as a binary representation of 1. This means that if we use three bits for representation of the solution, then the register **z** should be initialized as $|1\rangle|0\rangle|0\rangle$, because this state corresponds to the three-bit binary representation $001_2$ of the integer 1. Registers **a, b, c** and **t** are qubits for auxiliary computations and should be initialized as containing $|0\rangle$ states.

To understand how the circuit in Fig 3 can be decomposed into lower-level quantum gates, let's first revisit the idea which is used to construct the circuit of the interest. Using the property of modular multiplication,

$$(A \times B) \bmod N = ((A \bmod N) \times (B \bmod N)) \bmod N, \tag{1}$$

we can see that modular exponentiation is a succession of modular multiplications:

$$y^x \bmod N = (y^{x_0 2^0} \times y^{x_1 2^1} \times \ldots \times y^{x_{n-1} 2^{n-1}}) \bmod N =$$

$$= (\ldots ([(y^{x_0 2^0} \times y^{x_1 2^1}) \bmod N] \times \ldots \times y^{x_{n-1} 2^{n-1}}) \ldots \bmod N) \bmod N, \tag{2}$$

where $x = x_0 2^0 + x_1 2^1 + \ldots + x_{n-1} 2^{n-1}$. The above expression can be computed by successive multiplications modulo $N$ of 1 on $m_i(x_i) = y^{x_i 2^i}$, where $i$ goes from 0 to $n-1$. This multiplication is an operation controlled by $x_i$ : $m_i(1) = y^{2^i} \bmod N$ and $m_i(0) = 1$. We note that the values of $y^{2^i} \bmod N$ can be computed efficiently on a classical computer. Then modular multiplication operation can be represented by modular additions in the following way:

$$zm_i(x_i) \bmod N = (z_0 2^0 m_i(x_i) + z_1 2^1 m_i(x_i) + \ldots + z_{n-1} 2^{n-1} m_i(x_i)) \bmod N, \tag{3}$$

where $z = z_0 2^0 + z_1 2^1 + \ldots + z_{n-1} 2^{n-1}$ is an accumulated product at the $i$-th step. Finally, modular addition of two integers $A, B < N$ can be represented in the form

$$A + B \bmod N = \begin{cases} A + B, & \text{for } A + B < N; \\ A + B - N, & \text{for } A + B \geq N. \end{cases} \tag{4}$$

Let us make some comments on notations: dashed blue wires serve as auxiliary for lower-level operations. We decided to keep them in schemes in order for the reader not to lose track of what's going on. Circuit elements specified by precomputed classical values, namely $N$ and $m_i(1)$, shown by thick red lines.

Let us then build the circuit starting from the lowest level, with elementary quantum operations, and getting to the highest level of modular multiplications.

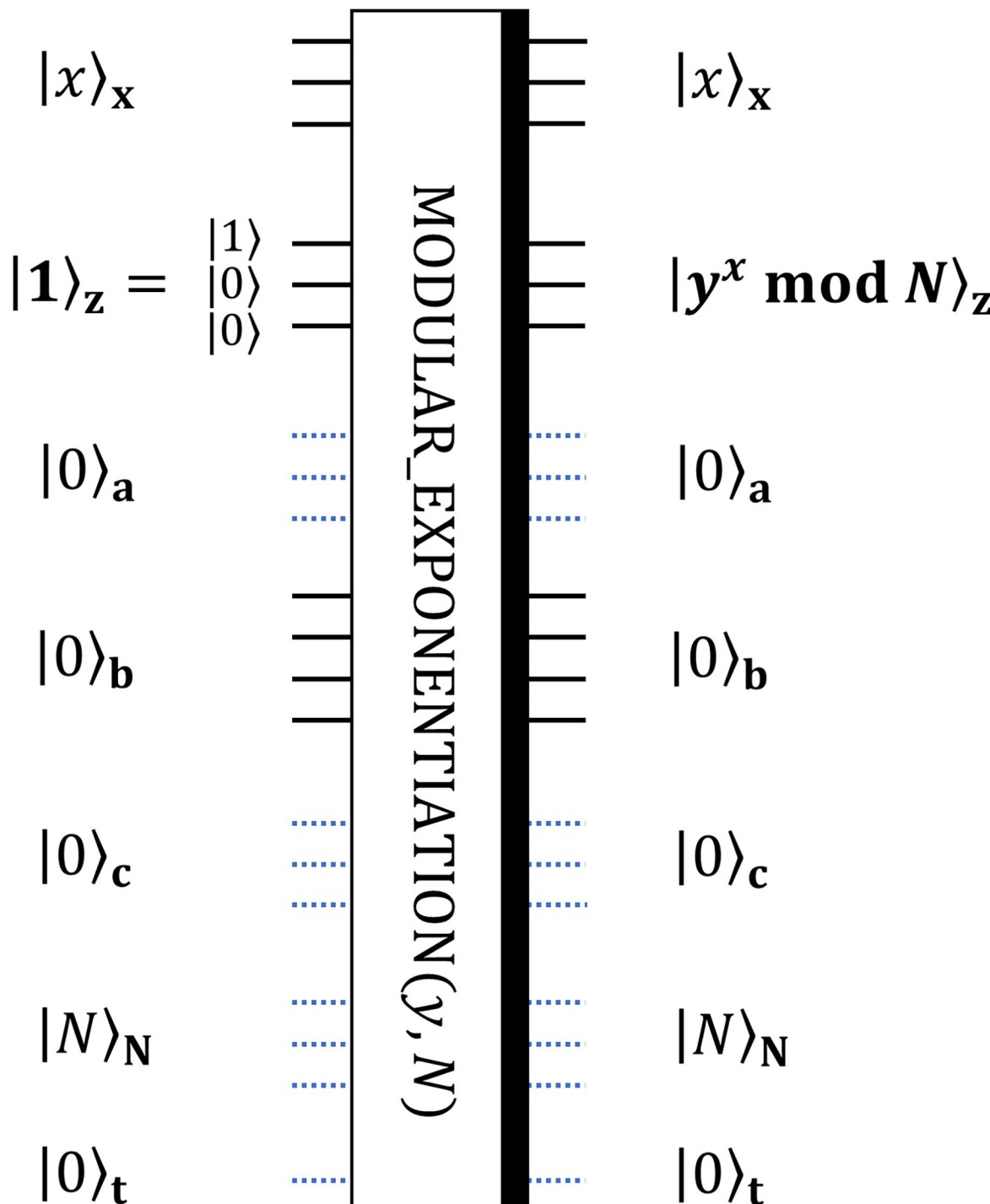

**Fig 3. General notation for the circuit realizing modular exponentiation procedure.** The procedure is for finding $y^x \bmod N$ given 3-bit integers $x$, $y$ and $N$.

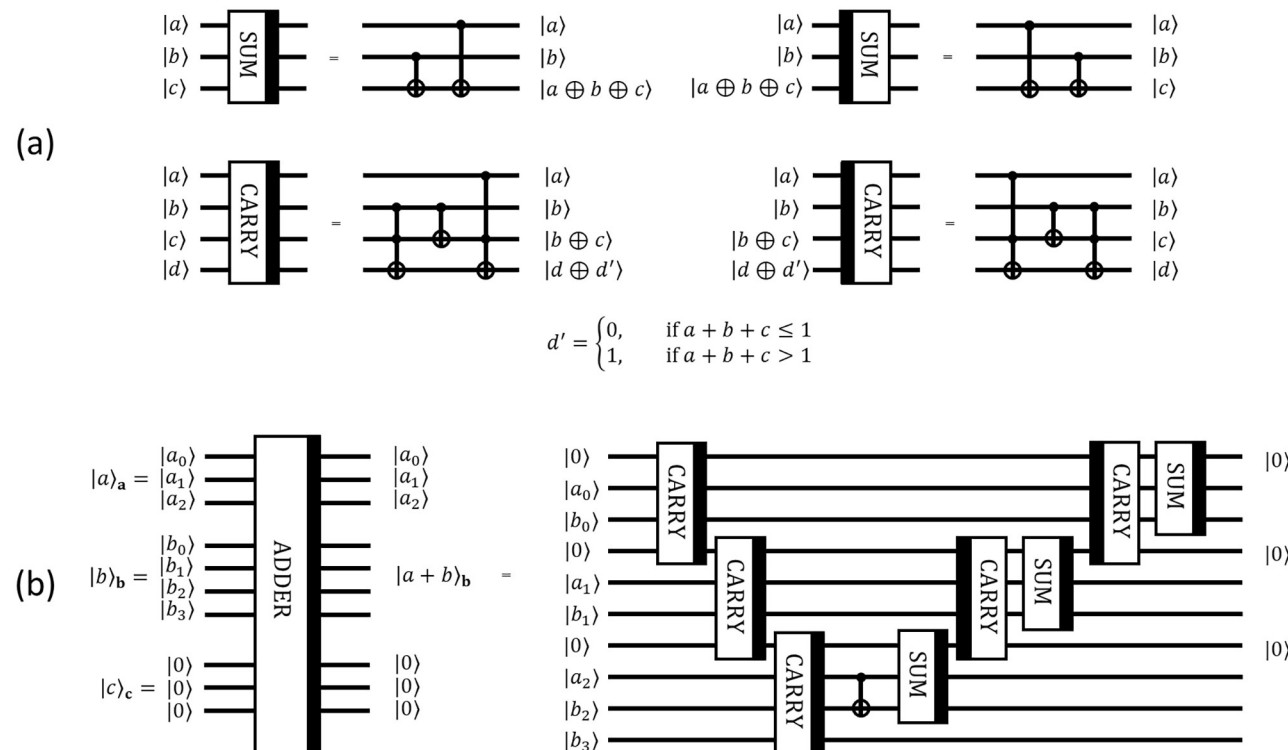

**Fig 4. SUM, CARRY and ADDER decompositions.** (a) Decomposition of SUM, reversed SUM, CARRY, and reversed CARRY circuits (b) Decomposition of ADDER circuit for adding two 3-bit integers $a$ and $b$.

**3.1 3-qubit addition circuit ADDER.** We use elementary circuits CARRY and SUM which implement bit-wise carry and sum operations. Their decompositions to CNOT and Toffoli gates are given in Fig 4(a). Note that a thick black line on the right side of a block denotes operation itself, while a thick black line on the left side of a block denotes a reversed (conjugate-transposed) operation, i.e. the operation with the reverse order of all elementary operations for the block with conjugation, if necessary. In fact, the reversed operation corresponds to a Hermitian conjugation of the initial operation.

The circuit SUM obtains a sum modulo 2 of two bits, while the idea of the circuit CARRY is to provide a 'carry bit' $\delta(a, b, c)$ corresponding to a standard summation of three bits $a, b, c \in \{0, 1\}$:

$$\delta(a, b, c) = \begin{cases} 0, & \text{if } a + b + c \leq 1; \\ 1, & \text{if } a + b + c > 1. \end{cases} \tag{5}$$

Then, CARRY and SUM are used to construct a 3-qubit addition transformation ADDER depicted in Fig 4(b). We note that here $a, b \in \mathbb{N}$ are numbers encoded by 3 (qu)bits, while the output register **b** contains an additional qubit to account for the possibility of a 4-bit result of the addition.

**3.2 3-qubit modular addition circuit ADDER_MOD.** Using ADDER and reversed ADDER circuits, we can construct modular addition by combining circuits in Block 1 and in Block 2, as shown in Fig 5. The idea behind the Block 1 is the following: firstly, ADDER performs the

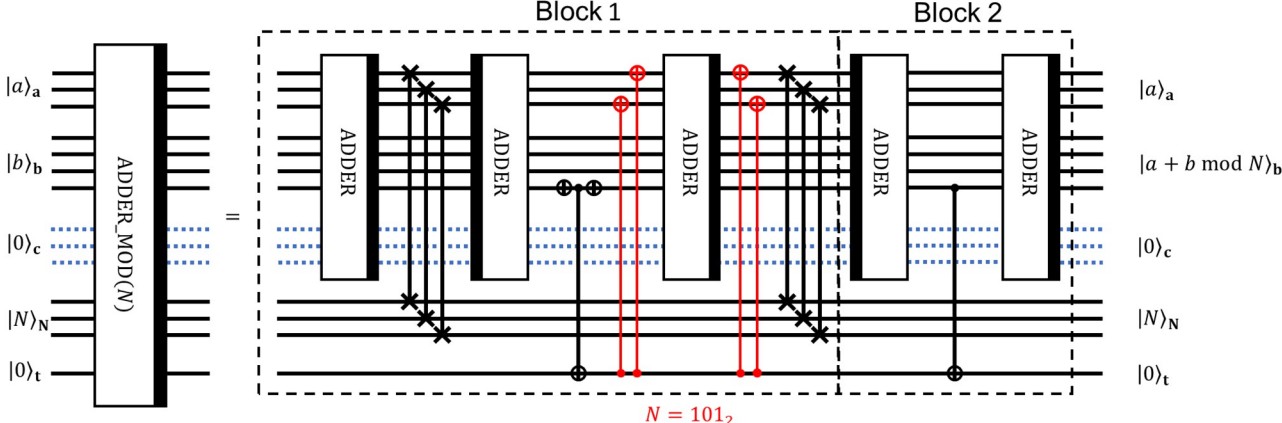

**Fig 5. Decomposition of `ADDER_MOD` circuit into lower-level operations.** The decomposition realizes modular addition of two 3-bit integers $a$ and $b$ modulo 3-bit integer $N$.

transformation

$$|a\rangle_{\mathbf{a}}|b\rangle_{\mathbf{b}}|0\rangle_{\mathbf{c}}|N\rangle_{\mathbf{N}}|0\rangle_{\mathbf{t}} \rightarrow |a\rangle_{\mathbf{a}}|a + b\rangle_{\mathbf{b}}|0\rangle_{\mathbf{c}}|N\rangle_{\mathbf{N}}|0\rangle_{\mathbf{t}}. \tag{6}$$

Then 3 SWAP gates swap the register **a** with the register **N** as follows:

$$|a\rangle_{\mathbf{a}}|a + b\rangle_{\mathbf{b}}|0\rangle_{\mathbf{c}}|N\rangle_{\mathbf{N}}|0\rangle_{\mathbf{t}} \rightarrow |N\rangle_{\mathbf{a}}|a + b\rangle_{\mathbf{b}}|0\rangle_{\mathbf{c}}|a\rangle_{\mathbf{N}}|0\rangle_{\mathbf{t}}. \tag{7}$$

An applying of the reversed `ADDER` results in the transformation

$$|N\rangle_{\mathbf{a}}|a + b\rangle_{\mathbf{b}}|0\rangle_{\mathbf{c}}|a\rangle_{\mathbf{N}}|0\rangle_{\mathbf{t}} \rightarrow$$
$$|N\rangle_{\mathbf{a}}|\gamma(a, b, N)\rangle_{\mathbf{b}}|0\rangle_{\mathbf{c}}|a\rangle_{\mathbf{N}}|0\rangle_{\mathbf{t}}, \tag{8}$$

where $\gamma(a, b, N) = a + b - N$ for $a + b - N \geq 0$ or $\gamma(a, b, N)$ is some bitstring with the higher order bit equal to 1 for $a + b - N < 0$.

The operation of the remaining part of the circuit is determined by the sign of $a + b - N$. If it is greater than 0, we want to keep the result in the register **b**, but if it is less than 0, we want to make an addition of $N$ once again to get $a + b$ in the register **b**. Recall that the information about the sign of $a + b - N$ is stored in the highest order bit of register **b**.

If it is equal to 0, then $a + b - N \geq 0$ and the register **b** already contains the $a + b \bmod N$. Using a `CNOT` gate with a target $t$, and then applying a number of `CNOT`s with control on $t$ leads to erasing the value of $N$ from the register **a** and replacing it by 0. Therefore, the third `ADDER` keeps the value of the register **b**. Then, we put back the value of $N$ in the register **a** and swap values $N$ and 0 between registers **a** and **N** to return registers **a** and **N** in the original state. Block 2 is applied to uncompute the value of register **t**.

In the case of $a + b - N < 0$, the third `ADDER` serves as inverse for the second one, thus, restoring the value of $a + b$ in the register **b**. `SWAP` operations set the initial values in registers **a** and **N**, and Block 2 is equivalent to the identity operator.

**3.3 3-qubit controlled modular multiplication circuit `Ctrl_MULT_MOD`.** The circuit `Ctrl_MULT_MOD` is given in Fig 6 and it implements a controlled modular multiplication of integers $z$ and $m$ modulo $N$ as a sequence of modular additions of integers $z_i 2^i \cdot m \bmod N$. The

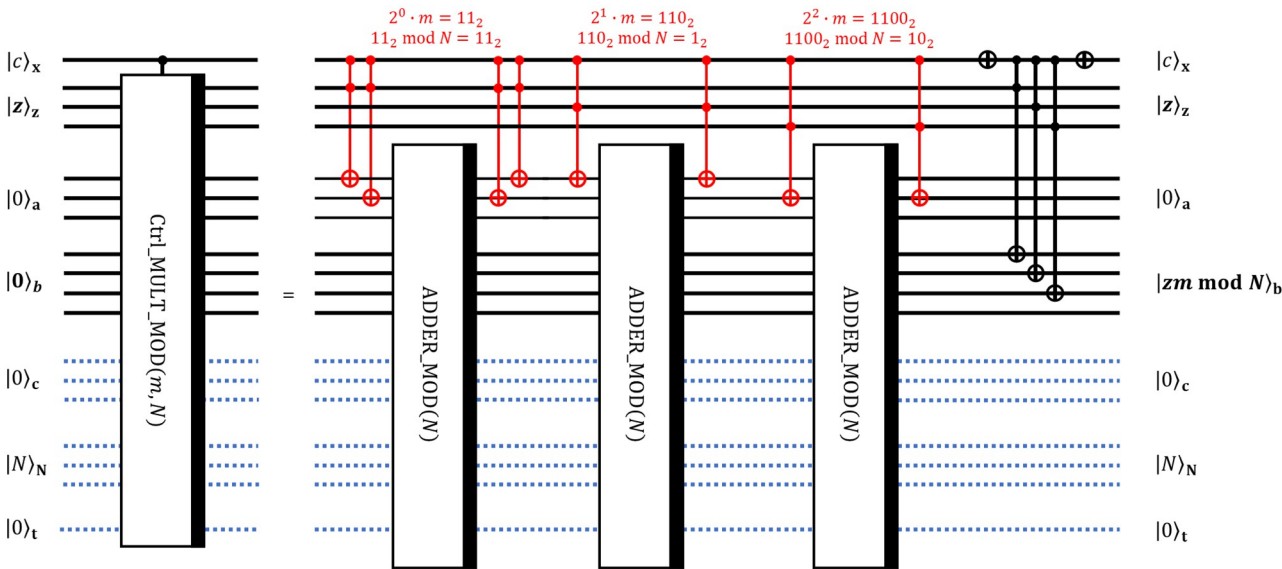

**Fig 6. Decomposition of `Ctrl_MULT_MOD` circuit into lower-level operations.** The decomposition realizes controlled modular multiplication of two 3-bit integers $z$ and $m$ modulo 3-bit integer $N$.

resulting transformation takes the form:

$$|c\rangle_x|z\rangle_z|0\rangle_a|0\rangle_b|0\rangle_c|N\rangle_N|0\rangle_t \rightarrow |c\rangle_x|z\rangle_z|0\rangle_a|zm \bmod N\rangle_b|0\rangle_c|N\rangle_N|0\rangle_t, \text{ if } c = 1$$

$$|c\rangle_x|z\rangle_z|0\rangle_a|0\rangle_b|0\rangle_c|N\rangle_N|0\rangle_t \rightarrow |c\rangle_x|z\rangle_z|0\rangle_a|z\rangle_b|0\rangle_c|N\rangle_N|0\rangle_t, \text{ if } c = 0$$

$$(9)$$

For this particular block we use $m = 3 = 11_2$, $N = 5 = 101_2$. The role of red `Toffoli` gates is to replace zeros in the register $|0\rangle_a$ with the state $|z_i 2^i \cdot m \bmod N\rangle_a$ to further add up all these numbers to get $|z \cdot m \bmod N\rangle_b$. Red `Toffoli` gates put values $2^i \cdot m \bmod N$ in the register **a** conditionally on values in registers **x** and **z**. We note that numbers $2^i \cdot m \bmod N$ can be efficiently computed on a classical computer. Also, note that this is the second time when classically precomputed information affects the configuration of the quantum circuit.

The last block of `CNOT`s is used to put the value $z$ in the register $|0\rangle_b$ if control $|c\rangle_x$ is $|0\rangle_x$.

**3.4 3-qubit modular exponentiation circuit `MODULAR_EXPONENTIATION`.** Finally, using an array of controlled modular multiplications, we can implement modular exponentiation using known classical information for every step as depicted in Fig 7. It should be a succession of controlled modular multiplications with controls set on wires of the register **x**. But every `Ctrl_MULT_MOD` should be accompanied by SWAPs and reversed `Ctrl_MULT_MOD` to reset one of the registers to zero and free it for the next controlled modular multiplication. The notation $(\ldots)^{-1} \bmod N$ is for modular inverse, which can be efficiently classically precomputed using Euclid's algorithm.

To sum up, `Ctrl_MULT_MOD` blocks implement the following chain of transformations which lead to the desired result:

$$|x\rangle_x|1\rangle_z|0\rangle_a|0\rangle_b|0\rangle_c|N\rangle_N|0\rangle_t \rightarrow$$

$$|x\rangle_x|1 \times y^{x_0 2^0} \bmod N\rangle_z|0\rangle_a|0\rangle_b|0\rangle_c|N\rangle_N|0\rangle_t \rightarrow$$

$$|x\rangle_x|1 \times y^{x_0 2^0} \times y^{x_1 2^1} \bmod N\rangle_z|0\rangle_a|0\rangle_b|0\rangle_c|N\rangle_N|0\rangle_t \rightarrow$$

$$(10)$$

$$\rightarrow \ldots \rightarrow |x\rangle_x|y^x \bmod N\rangle_z|0\rangle_a|0\rangle_b|0\rangle_c|N\rangle_N|0\rangle_t$$

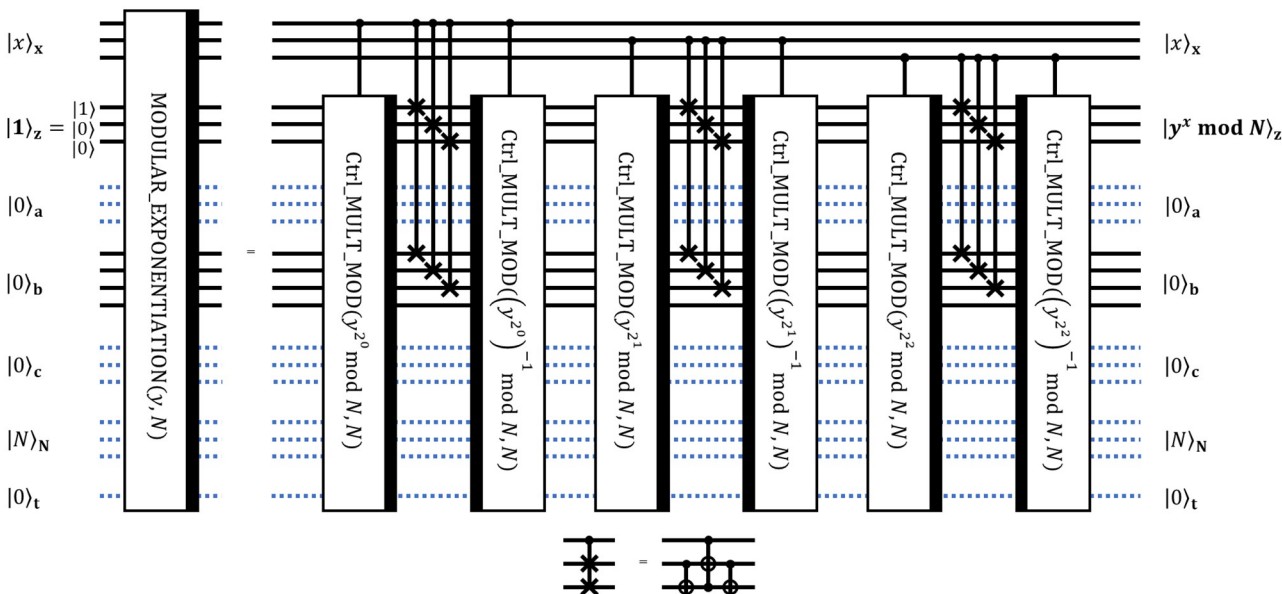

**Fig 7. Decomposition of `MODULAR_EXPONENTIATION` circuit into lower-level operations.** The decomposition realizes modular exponentiation $y^x$ mod $N$ given 3-bit integers $x$, $y$, and $N$.

It is worth mentioning that if the size of the register $|N\rangle_N$ is $n$, then the size of the register $|x\rangle_x$ should be greater than or equal to $2n+1$ to make `MODULAR_EXPONENTIATION` circuit usable in Shor's algorithm (see [16]). For instance, going to $2n + 2 = 8$ qubits in $|x\rangle_x$ for this particular case requires just additional 5 wires for $|x\rangle$ and additional 5 blocks of [Ctrl_MULT_MOD—SWAPs—reversed `Ctrl_MULT_MOD`] in Fig 7.

Lastly, let us consider the situation when we increase integers for which we want to compute modular exponentiation. If we go from 3-bit integers to 4-bit integers, then the current architecture requires 4 qubits for each of registers **x**, **z**, **a**, **c**, and **N**; 4+1 qubits for the register **b**; and 1 qubit for control **t**. Thus, one can see that the number of qubits grows as $O(n)$, which is acceptable according to the original paper.

## 4 Example

Here we provide an example of usage of the template `Order_Finding` inside PennyLane standard environment. The script from the Listing 3 is designed to find with high probability the least positive integer $r$ such that $y^r$ mod $N = 1$ for 3-bit integers $y = 3$ and $N = 5$. The size of the register **x** is $2n + 2 = 2\cdot3 + 2 = 8$.

**Listing 3.** Example of usage of the template `Order_Finding`.

```
import pennylane as qml
import QuantumOperations as q
# define initial parameters
N = 5
y = 3
bits_for_register_with_a_number = 3
bits_for_x_register = 2*bits_for_register_with_a_number + 2
# define wires with all registers
wires=[i for i in range(bits_for_x_register+bits_for_register_with_
a_number*5+2)]
# device
```

```
dev = qml.device('default.qubit', wires = wires, shots = 10000,
analytic = None)
# circuit
def func(N, y,bits_for_x_register,input_):
  # insert input
  for i in range(len(wires)):
    if input_[i] == 1:
      qml.PauliX(wires = wires[i])
  # circuit
  q.Order_Finding(N, y,bits_for_x_register,wires = wires)
  return qml.probs(wires=[0, 1, 2, 3, 4, 5, 6, 7])
# QNode
circuit = qml.QNode(func,dev)
# Run calculations for given parameters with the
# register wires_N initialized as binary N and
# register wires_z – as binary 1
measurements_probabilities = circuit (5, 3,bits_for_x_register, [0,
0, 0, 0, 0, 0, 0, 0] + [1, 0, 0] + [0, 0, 0] + [0, 0, 0, 0] + [0, 0, 0]
+ [1, 0, 1] + [0])
```

Results of measurements in the constructed circuit are put into the variable "measurements_probabilities" as an array. After post-processing, we can get the probability distribution of measurements as depicted in Fig 8. Each bar corresponds to a particular measurement outcome that can be interpreted as an estimate of $s/r$, where $r$ is the order of $y$ modulo $N$ and $s$ is some integer.

In particular, if measurement has the form $|x_1 x_2 \ldots\rangle$, then the estimate of $s/r$ is the number $0.x_1 x_2 \ldots$ in the binary representation, and possible values of $r$ can be reconstructed from this estimate. According to the algorithm, measurements with high probability correspond to estimates that are close to the true value of $s/r$.

In our case, the four measurements with the greatest probabilities are $|000000\rangle$, $|100000\rangle$, $|111111\rangle$ and $|010000\rangle$. These measurements correspond to representations of $s/r$ in the form $0.000000 = 0$, $0.100000 = 1/2$, $0.111111 = 63/64$ and $0.010000 = 1/4$, respectively. It can be seen that the fourth result gives a proper value of $r$, since $y^r \bmod N = 3^4 \bmod 5 = 1$.

## 5 Transpilation and resource-estimation

For real-life implementations of the given algorithm it is important to translate the decomposition into a set of gates that are native for a platform of the interest. This process of translation is called transpilation, and for many frameworks particular details of this process are not explained to a user. Although results of transpilation oftentimes can be accessed, it is not clear to which extent those results can be reliable for estimation of resources for an algorithm, such as non-Clifford gate count and execution time expressed in the depth of a transpiled algorithm.

The advantage of using PennyLane package for the realization of the algorithm is the ability to run algorithms on different platforms. It allows for direct comparison of algorithms' performance for different platforms, which itself can be a subject of research (see, for instance, [17]). With increase in hardware's computing capabilities, it will be harder to compile algorithms for different platforms, so unified framework such as PennyLane library might provide both tools for realizing algorithms and transparency in rules of decomposing these algorithms to the level of native gates.

To illustrate this reasoning, we present a simplified protocol for transpilation which is derived from [18] and provide a table with upper bounds on gate counts and depth of the order-finding algorithm. The table was derived by direct application of the transpilation protocol that we realized using PennyLane library.

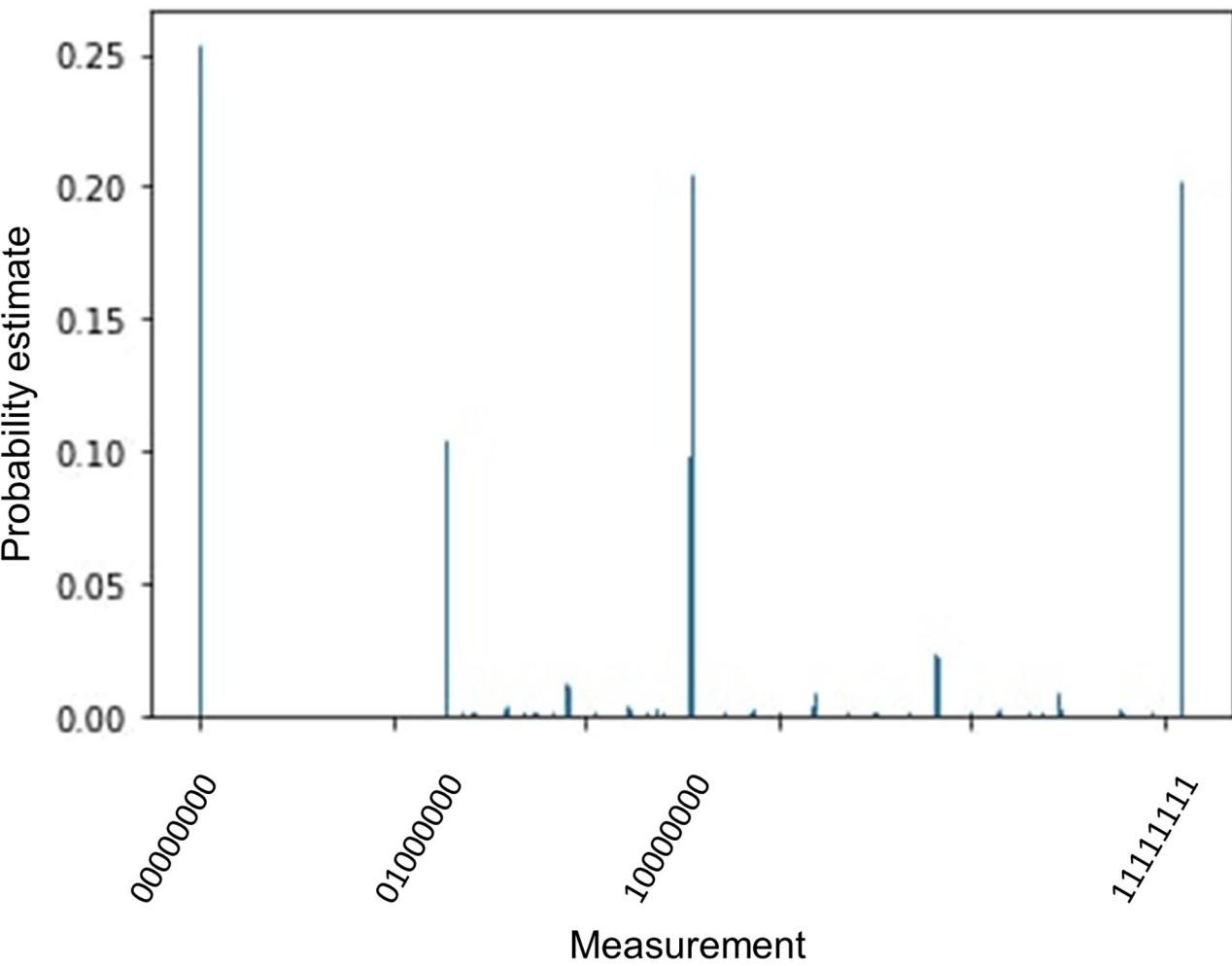

**Fig 8. Probabilities of measurements for order finding procedure.**

Additionally, the original work [18] lacks analytic expressions that are necessary for an efficient implementation of single-qubit decompositions and lacks the proof of universality of the single-qubit unitary operation decomposition. We provide these analytic expressions in the next subsection and give proofs in S2 Appendix.

An open question for further development of this work is how to define criteria for choice of algorithms for realization. Shor's algorithm has many different protocols of realizations with various advantages and disadvantages, and the protocol of the realization in our work was chosen for two reasons: efficiency of realization in terms of gate counts and simplicity of exposition. Realization of transpilation techniques could be more sophisticated as well, and it remains unclear which particular algorithms will be of greater interest in the future.

**5.1 Native gates for trapped-ion qubits.**   Native single-qubit gate with unitary evolution operator

$$R(\theta, \phi) = \begin{pmatrix} \cos\dfrac{\theta}{2} & -ie^{-i\phi}\sin\dfrac{\theta}{2} \\ -ie^{i\phi}\sin\dfrac{\theta}{2} & \cos\dfrac{\theta}{2} \end{pmatrix}, \tag{11}$$

and native two-qubit gate with unitary evolution operator

$$XX(\chi) = \begin{pmatrix} \cos\chi & 0 & 0 & -i\sin\chi \\ 0 & \cos\chi & -i\sin\chi & 0 \\ 0 & -i\sin\chi & \cos\chi & 0 \\ -i\sin\chi & 0 & 0 & \cos\chi \end{pmatrix} \tag{12}$$

are used in trapped-ion quantum computer [18]. Available sign of $\chi$ is defined by characteristics of particular experimental tool [18, 19]. For ease of exposition, we assume that this sign is positive for each pair of qubits, although arbitrary signs can be easily introduced in decompositions as an input parameter.

Note that an arbitrary unitary operation $U$ can be decomposed into a sequence of at most two native single-qubit gates [18]

$$U = \begin{pmatrix} u_{00} & u_{01} \\ u_{10} & u_{11} \end{pmatrix} = e^{id}R(-\pi, -c - \pi/2)R(2b + \pi, a - c - \pi/2), \tag{13}$$

where $u_{ij}$ are complex elements of matrix $U$, and $a$, $b$, $c$, $d$ are real parameters. The proof of this result and analytic expressions for $a$, $b$, $c$ was not considered in the original work, although these expressions are crucial for effective decomposition of single-qubit gates. These expressions are

$$a = \frac{1}{2}(\varphi_{00} - \varphi_{11}), \quad b = \arccos|u_{00}|, \quad c = \frac{1}{2}(\varphi_{00} - 2\varphi_{10} + \varphi_{11}) - \pi, \tag{14}$$

where $\varphi_{ij} = \mathrm{Arg}(u_{ij})$. Proofs can be found in S2 Appendix.

**5.2 Simplified transpilation protocol.** The protocol borrows the simplest steps 1–4 as well as combining single-qubit gates from the last step of the protocol given in [18]. It can be briefly formulated as the following sequence of steps:

1. Translate all operations into set {3-qubit Toffoli, CNOTs, single-qubit operations}.

2. Translate 3-qubit Toffoli to Controlled-V and CNOTs, where Controlled-V represents controlled square-root-of-$X$ operation (see [20]).

3. Translate Controlled-V and CNOTs into set {XX, single-qubit operations}.

4. For every set of concurrent single-qubit gates, translate this set into one resulting operation and decompose it to at most 2 rotations $R$.

This protocol does not include possibility to bind operations into blocks that can be executed simultaneously. Since estimation of execution time might be significantly affected by parallelization of the circuit operations, we developed a simple algorithm to estimate depth of the circuit.

By construction, at the end of the transpilation there are at most 2 single-qubit $R$-gates per one qubit between any pair of two-qubit gates. Thus, the estimate of the circuit's depth with only two-qubit gates multiplied by 3 will give an upper bound.

The counting algorithm has the following steps:

1. Exclude all single-qubit operations from the list of transpilled operations.

2. Prescribe number '0' to every qubit.

**Table 3. Resource estimation for Shor's algorithm on trapped-ion platform.**

| Maximal value of $N$ | All native operations | Two-qubit native operations | Depth |
|---|---|---|---|
| $2^2$ | 23941 | 5010 | 3808·3 |
| $2^3$ | 77054 | 16152 | 11440·3 |
| $2^4$ | 174649 | 36650 | 25648·3 |
| $2^5$ | 340520 | 71452 | 48615·3 |

3. Iteratively take a two-qubit gate from the list of transpilled operations and update numbers prescribed to the qubits involved in the current two-qubit operation. In particular, prescribe number '$m + 1$' to the two qubits, where $m$ is the maximal number prescribed to the two qubits during previous iterations.

4. Find the maximal prescribed number among all qubits. This number multiplied by 3 is equal to the upper bound on depth of the circuit.

**5.3 Resource estimation.** Table 3 represents counts of native gates and depths for the realization of Shor's algorithm using the simplified transpilation procedure.

Maximal values of $N$ were chosen in the form $2^n$, because the change in $n$ represents the change in the size of qubit register. For a fixed size of qubit register, there is no significant change in the number of operations across different values of $N$.

## Conclusion

In the present work, we have shown a package based on the PennyLane library implementing decompositions to elementary quantum gates all blocks of the quantum parts of Shor's algorithm and further transpilation of the decomposition to the level of native operations for ion-trapped quantum processor. Current realization can be built into the PennyLane library as quantum gates and can be used for experiments on quantum computers and quantum simulators, as well as for resource estimation before running an algorithm. We hope that combination of realization and study of aspects of the implementation represents interesting contribution to scientific community. Our study shows that still there is a gap between known academic results in the field of quantum information theory and implementation of quantum algorithms using currently available quantum platforms. For example, our idea to use results of Maslov [18] was not directly realizable and additional research on the universality of single qubit decomposition and derivation of decomposition's coefficients were needed. We expect that our developments will be used as pre-prorgrammed primitives for a broader range of quantum algorithms.

## Supporting information

**S1 Appendix. Shor's algorithm [16].**
(PDF)

**S2 Appendix. Single-qubit unitary decomposition: Analytic expressions.**
(PDF)

## Author Contributions

**Conceptualization:** A. V. Antipov, A. K. Fedorov.

**Formal analysis:** A. V. Antipov, E. O. Kiktenko.

**Funding acquisition:** E. O. Kiktenko, A. K. Fedorov.

**Investigation:** A. V. Antipov.

**Methodology:** A. V. Antipov, E. O. Kiktenko.

**Project administration:** A. K. Fedorov.

**Resources:** A. K. Fedorov.

**Software:** A. V. Antipov.

**Supervision:** E. O. Kiktenko, A. K. Fedorov.

**Validation:** E. O. Kiktenko, A. K. Fedorov.

**Visualization:** A. V. Antipov.

**Writing – original draft:** A. V. Antipov.

**Writing – review & editing:** E. O. Kiktenko, A. K. Fedorov.

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
