## [Decision Letter · Decision Letter 0]

10 Apr 2022

PONE-D-22-06127Efficient realization of quantum primitives for Shor’s algorithm using PennyLane libraryPLOS ONE

Dear Dr. Antipov,

Thank you for submitting your manuscript to PLOS ONE. After careful consideration, we feel that it has merit but does not fully meet PLOS ONE’s publication criteria as it currently stands. Therefore, we invite you to submit a revised version of the manuscript that addresses the points raised during the review process.

In particular, at least, results of the developed example should be discussed in more detail (including a solved problem of higher complexity), and a critical assessment of the advantage of using the proposed library and further examples of problems suitable to be targeted should be given.

We look forward to receiving your revised manuscript.

Kind regards,

Pietro Massignan, PhD

Academic Editor

PLOS ONE

Journal Requirements:

2. Please upload a copy of Supporting Information which you refer to in your text on page 5.

Reviewers' comments:

Reviewer's Responses to Questions

**Comments to the Author**

1. Is the manuscript technically sound, and do the data support the conclusions?

Reviewer #1: Partly

Reviewer #2: Partly

2. Has the statistical analysis been performed appropriately and rigorously? 

Reviewer #1: N/A

Reviewer #2: N/A

3. Have the authors made all data underlying the findings in their manuscript fully available?

Reviewer #1: Yes

Reviewer #2: Yes

4. Is the manuscript presented in an intelligible fashion and written in standard English?

Reviewer #1: Yes

Reviewer #2: Yes

5. Review Comments to the Author

Reviewer #1: Dear authors,

I found the idea in this paper a useful tool to be added to a software library. The package here provided by the authors can be used in the future to implement Shor's algorithm easily. However, I cannot recommend this work for publication due to the lack of originality it has.

The first requirement that the journal demands for a paper to be published is that the research must be original. This is not fulfilled since the package here presented is a direct implementation of the gates and operations described in Ref. [13], which is a 26 years old paper. No further improvement has been added, and even the figures are really similar. On the software leg, a significant contribution on computational techniques could support the publication, but this work only constructs a package on a pre-existent library and wraps the computational flow so that final results are directly obtained.

On the side of data availability, the package is open-source and can be easily accessed. No further information is needed to reproduce the given algorithm. English and writing is also good, only a couple of informal sentences should be modified.

In my opinion, this work could better suit a hackathon or a open call for packages in the aforementioned library, in this case PennyLane, than rather a scientific publication. I encourage the authors to get in contact with PennyLane and upload this package to their platform.

Reviewer #2: The manuscript by A.Antipov, E.Kiktenko and A.Fedorov presents a software package that contains implementations of building blocks and quantum gates for the realization of quantum algorithms using PennyLane, a software platform that uses Python for programming quantum computers. In particular, the templates included in this package allow to implement Shor’s algorithm, specifically the part that needs quantum computation (the order-finding procedure). The main work is clearly stated: the authors aim to present a set of functions that allow to implement Shor’s algorithm using PennyLane library. Furthermore, they make available the files containing the programmed functions for PennyLane in Github, which I really appreciate, because it is always useful to have a clear and direct access to additional information and resources besides all the materials and sources.

The paper is easy to follow because it is good structured and organised. It first offers a general overlook of the package and states its main contents (quantum circuits realized as PennyLane’s templates and classical functions for auxiliary computations), while providing a complete and efficiently summed up explanation of the full list of developed modules/templates and classical functions [Table 1]. The authors describe the order finding quantum procedure that is necessary for the realization of Shor’s algorithm. They make good use of an appendix for presenting a quick overview of Shor's algorithm, which acts as a reminder for the reader. The document includes an illustrative circuit [Fig. 2] of the order-finding procedure (consisting of Hadamard gates, a modular exponentiation and an inverse quantum Fourier transform) and makes use of this resource as a complementary tool for facilitating the understanding of the theorical procedure. The quantum modular exponentiation is the most important building block of the order-finding procedure, so the authors present a fully description of this method for 3-bit integer inputs as an example (although the inputs can be n-bits long). They first state a mathematical explanation of how the modular exponentiation can be decomposed. Since it is a complex process, they decomposed it into lower-level quantum gates: the modular exponentiation is a succession of modular multiplications; which can be represented by modular additions, which in turn make use of just bit-wise sum and carry operations. Hence the main circuit is broken into easier parts and each one is analysed separately. The manuscript ends up offering an illustrative example of the order-finding procedure, which I personally appreciate. The authors achieve to connect their work with published literature, since their templates and functions can be used for complementing existing works and developing new quantum algorithms.

The whole of the text is well-written and easy to follow, since all the followed steps are well explained and addressed. A very useful resource is to combine the theoretical explanation with the computational code [Listings] and the scheme of the circuit implementing this functionality. The combination of these three factors results in a crucial and successful tool for making the article accessible to the vast majority of a public minimally acquainted with quantum algorithms and Python language. The visual resources such as tables, coding and circuit schemes serve as a complement for consolidating the understanding of what is being asserted. Specially, a really effective point is the pairing of the schematic circuit with its analogue coding implementation, since they complement each other. This connection allows a correct comprehension both of the programming and the functionality of the circuit, so that the redundancy acts as a strength mechanism for reinforcing the reader's understanding. Also, the use of these resources enables the authors to support their claims, since they provide evidence for the authors’ main findings and developments. Therefore, I recommend this manuscript to be published in PLOS ONE.

Here below follows a list of suggestions and comments, which the authors may want to take into account before resubmission:

- The main issue is that the article fails to efficiently justify the relevance of its application. Because surely Shor’s algorithm can be implemented using other software platforms. So which is the major contribution and advantage of using the package in PennyLane? What makes the authors’ work appealing and why is it innovative, original and significant? I cannot find a direct and clear answer to these questions. Maybe it could help to specify and discuss the innovative reasons that lead the authors to create such templates.

- It would be interesting to present the limitations of the work, as well as some further applications or suggest some open questions or future problems to address. Maybe to aim or point at creating new blocks or functions, or find more algorithms to implement using PennyLane software.

- Given that an example is provided [Sec.4], it could be more developed. The obtention of the final result should be more clearly explained, since this data analysis would definitely provide evidence for the stated conclusions. The results concluded from the graph of probabilities [Fig. 8] are not commented, so it would be useful to include a discussion of the final results.

- [Table 2] shows a list of functions for auxiliary computations, but neither of these classical functions is used as an auxiliary one in the templates presented in the document, and they are not even mentioned. Maybe it would be nice to state where and how these functions are being used and for what purpose.

6. PLOS authors have the option to publish the peer review history of their article (what does this mean?). If published, this will include your full peer review and any attached files.

Reviewer #1: No

Reviewer #2: No

---

## [Author Response · Author response to Decision Letter 0]

8 Jun 2022

Dear reviewers, 

I provide the full text of the uploaded file 'Response to Reviewers' below.

We appreciate your efforts in arranging the peer review of our manuscript. We are also grateful to the Reviewers for useful suggestions and insightful comments that could help us to improve our work. We have prepared a revised version of our manuscript; we have highlighted all the changes.

Please see point-by-point answers to Reviewers below.

Reviewer #1

Question/Comment: The first requirement that the journal demands for a paper to be published is that the research must be original. This is not fulfilled since the package here presented is a direct implementation of the gates and operations described in Ref. [13], which is a 26 years old paper. No further improvement has been added, and even the figures are really similar. On the software leg, a significant contribution on computational techniques could support the publication, but this work only constructs a package on a pre-existent library and wraps the computational flow so that final results are directly obtained.

Answer: First of all, we would like to thank the Reviewer for careful reading of our manuscript and overall assessment of our work. We made some improvements, in particular, we used a simplified technique for decomposition of algorithms into a set of gates which are native for trapped-ion quantum processor and realized this technique using PennyLane library. The decomposition is used to analyze resources required for an execution of Shor's algorithm on the level of native operations of trapped-ion quantum computer in Section 5. Our original contribution is the derivation of coefficients needed for implementation of the decomposition of single-qubit operations into native rotation operations as well as proof of universality of this decomposition. These results were not given in the original work [20] containing decomposition protocol. Overall, our study shows that there is a gap between known academic results in the field of quantum information theory and implementation of quantum algorithms on available quantum platforms. And we hope that combination of realization and study of aspects of the implementation represents interesting contribution to scientific community.

Question/Comment: English and writing is also good, only a couple of informal sentences should be modified

Answer: We would be happy to edit these sentences, but we would like to know what sentences in particular did the Reviewer mention.

Question/Comment: I encourage the authors to get in contact with PennyLane and upload this package to their platform.

Answer: We’ve been in contact with PennyLane’s team, and received a recommendation to “open a PR with the contribution … It will then follow the same process that any internal contribution would, and if as you say the benefit is evident and the necessary changes are manageable, then I see no reason why this shouldn't end up in a merge”. We are going to follow this recommendation as soon as possible.

Reviewer #2

Question/Comment: The main issue is that the article fails to efficiently justify the relevance of its application. Because surely Shor’s algorithm can be implemented using other software platforms. So which is the major contribution and advantage of using the package in PennyLane? What makes the authors’ work appealing and why is it innovative, original and significant? I cannot find a direct and clear answer to these questions. Maybe it could help to specify and discuss the innovative reasons that lead the authors to create such templates

Answer: We hope that Section 5 that we have added recently will be helpful to address concerns of the Reviewer. In this section, we used a simplified technique for decomposition of algorithms into a set of gates which are native for trapped-ion quantum processor and realized this technique using PennyLane library. The decomposition is used to analyze resources required for an execution of Shor's algorithm on the level of native operations of trapped-ion quantum computer. Our original contribution is the derivation of coefficients needed for implementation of the decomposition of single-qubit operations into native rotation operations as well as proof of universality of this decomposition. These results were not given in the original work [20] containing decomposition protocol. Overall, our study shows that there is a gap between known academic results in the field of quantum information theory and implementation of quantum algorithms on available quantum platforms. And we hope that combination of realization and study of aspects of the implementation represents interesting contribution to scientific community. Additionally, the choice of PennyLane library might be supported by the ability to work with different physical realizations of quantum processors within unified framework.

Question/Comment: It would be interesting to present the limitations of the work, as well as some further applications or suggest some open questions or future problems to address. Maybe to aim or point at creating new blocks or functions, or find more algorithms to implement using PennyLane software.

Answer: An open question for further development of this work is how to define criteria for choice of algorithms for realization. Shor's algorithm has many different protocols of realizations with various advantages and disadvantages, and the protocol of the realization in our work was chosen for two reasons: efficiency of realization in terms of gate counts and simplicity of exposition. Realization of transpilation techniques could be more sophisticated as well, and it remains unclear which particular algorithms will be of greater interest in the future. We added this comment in the manuscript.

Question/Comment: Given that an example is provided [Sec.4], it could be more developed. The obtention of the final result should be more clearly explained, since this data analysis would definitely provide evidence for the stated conclusions. The results concluded from the graph of probabilities [Fig. 8] are not commented, so it would be useful to include a discussion of the final results.

Answer: Following the advice of the Reviewer, we added more elaborate explanation of the final result in the manuscript.

Question/Comment: [Table 2] shows a list of functions for auxiliary computations, but neither of these classical functions is used as an auxiliary one in the templates presented in the document, and they are not even mentioned. Maybe it would be nice to state where and how these functions are being used and for what purpose.

Answer: Classical function modular_multiplicative_inverse is crucial for building the decomposition and it is used as the auxiliary function for MODULAR_EXPONENTIATION. The role of this classical function is to find parameters for the lower-level decomposition Ctrl_MULT_MOD_inv. Important aspect of modular_multiplicative_inverse is the efficiency of the realization that relies on the efficiency of two other classical functions gcd and diophantine_equation. We added this comment in the manuscript.

Question/Comment: Specially, a really effective point is the pairing of the schematic circuit with its analogue coding implementation, since they complement each other. This connection allows a correct comprehension both of the programming and the functionality of the circuit, so that the redundancy acts as a strength mechanism for reinforcing the reader's understanding. Also, the use of these resources enables the authors to support their claims, since they provide evidence for the authors’ main findings and developments. Therefore, I recommend this manuscript to be published in PLOS ONE.

Answer: We are grateful to the Reviewer for the appreciation of our work!

With kind regards,

The authors.

---

## [Decision Letter · Decision Letter 1]

1 Jul 2022

Efficient realization of quantum primitives for Shor’s algorithm using PennyLane library

PONE-D-22-06127R1

Dear Dr. Antipov,

We’re pleased to inform you that your manuscript has been judged scientifically suitable for publication and will be formally accepted for publication once it meets all outstanding technical requirements.

Kind regards,

Pietro Massignan, PhD

Academic Editor

PLOS ONE

Reviewers' comments:

Reviewer's Responses to Questions

**Comments to the Author**

1. If the authors have adequately addressed your comments raised in a previous round of review and you feel that this manuscript is now acceptable for publication, you may indicate that here to bypass the “Comments to the Author” section, enter your conflict of interest statement in the “Confidential to Editor” section, and submit your "Accept" recommendation.

Reviewer #1: All comments have been addressed

Reviewer #2: All comments have been addressed

2. Is the manuscript technically sound, and do the data support the conclusions?

Reviewer #1: Yes

Reviewer #2: Yes

3. Has the statistical analysis been performed appropriately and rigorously? 

Reviewer #1: N/A

Reviewer #2: N/A

4. Have the authors made all data underlying the findings in their manuscript fully available?

Reviewer #1: Yes

Reviewer #2: Yes

5. Is the manuscript presented in an intelligible fashion and written in standard English?

Reviewer #1: Yes

Reviewer #2: Yes

6. Review Comments to the Author

Reviewer #1: Dear authors,

thanks for addressing all the comments I suggested in the previous revision. Now it is possible to say that there is some original contribution in the paper. This is, however, somewhat straightforward, but as the authors mention it bridges the gap between theory and an experimental implementation, at least from the theoretical point of view.

The estimation of resources is an interesting calculation. However, I have the feeling that these numbers can be further reduced with an appropriate non-trivial compilation scheme. Do you have any comments on this? Have you explored this path? It is known that finding the optimal compilation and execution is a NP-hard problem, but maybe there is some sub-optimal configuration that is easily achievable and significantly reduces the computational cost.

About informal comments, this is my personal opinion, and as long as neither the editor nor the other referee mentioned it, please do not take it into consideration. Just as an example, in lines 222 - 225 the way of writing the conditions (we want, we want) sounds informal to me.

Reviewer #2: - The authors have successfully included a specific section for a real life application of their work, regarding a trapped-ion quantum processor. In order to do that, they introduce the concept of transpilation and they offer an explanation of this technique as a decomposition of a given algorithm into a set of gates that are native for a particular implementation of interest. The authors claim the importance of their PennyLane library by stating that it allows to realize an algorithm and decompose it into its native gates. The inclusion of a protocol for transpilation using PennyLane library is useful for complementing the theoretical reasoning.

- In this reviewed version of the paper, the authors include a development of the protocols of transpilation for Shor’s algorithm. They have also included a possible extension of their work by stating that the transpilation technique can be applied to other algorithms besides Shor’s one.

- The authors have included a more elaborated explanation of how to interpret the final results obtained in [Fig. 8]. Hence now the analysis is more understandable and consistent.

- As a last comment, the authors explain the role of the classical auxiliary functions as clue parts in the modular exponentiation process.

The authors have successfully achieved to address all the initial comments and have taken into account the reviewer’s considerations. Therefore, I recommend this manuscript to be published in PLOS ONE.

7. PLOS authors have the option to publish the peer review history of their article (what does this mean?). If published, this will include your full peer review and any attached files.

Reviewer #1: No

Reviewer #2: No

---

## [Editor Report · Acceptance letter]

6 Jul 2022

PONE-D-22-06127R1 

Efficient realization of quantum primitives for Shor’s algorithm using PennyLane library 

Dear Dr. Antipov:

I'm pleased to inform you that your manuscript has been deemed suitable for publication in PLOS ONE. Congratulations! Your manuscript is now with our production department. 

Kind regards, 

on behalf of

Dr. Pietro Massignan 

Academic Editor

PLOS ONE